# Impact of Bioinformatics Search Parameters for Peptides’ Identification and Their Post-Translational Modifications: A Case Study of Proteolysed Gelatines from Beef, Pork, and Fish

**DOI:** 10.3390/foods12132524

**Published:** 2023-06-28

**Authors:** Mouna Ambli, Barbara Deracinois, Anne-Sophie Jenequin, Rozenn Ravallec, Benoit Cudennec, Christophe Flahaut

**Affiliations:** UMR Transfrontalière BioEcoAgro-INRAe N° 1158, Univ. Artois, Univ. Lille, INRAe, Univ. Liège, UPJV, JUNIA, Univ. Littoral Côte d’Opale, ICV-Institut Charles Viollette, 62300 Lens, France; mouna.ambli@gmail.com (M.A.); barbara.deracinois@univ-lille.fr (B.D.); annesophie.jenequin@univ-lille.fr (A.-S.J.); rozenn.ravallec@univ-lille.fr (R.R.); benoit.cudennec@univ-lille.fr (B.C.)

**Keywords:** bioinformatics, gelatine, hydrolysate, hydroxyproline, mass spectrometry, post-translational modifications

## Abstract

Bioinformatics software, allowing the identification of peptides by the comparison of peptide fragmentation spectra obtained by mass spectrometry versus targeted databases or directly by de novo sequencing, is now mandatory in peptidomics/proteomics approaches. Programming the identification software requires specifying, among other things, the mass measurement accuracy of the instrument and the digestion enzyme used with the number of missed cleavages allowed. Moreover, these software algorithms are able to identify a large number of post-translational modifications (PTMs). However, peptide and PTM identifications are challenging in the agrofood field due to non-specific cleavage sites of physiological- or food-grade enzymes and the number and location of PTMs. In this study, we show the importance of customized software programming to obtain a better peptide and PTM identification rate in the agrofood field. A gelatine product and one industrial gelatine hydrolysate from three different sources (beef, pork, and fish), each digested by simulated gastrointestinal digestion, MS-grade trypsin, or both, were used to perform the comparisons. Two main points are illustrated: (i) the impact of the set-up of specific enzyme versus no specific enzyme use and (ii) the impact of a maximum of six PTMs allowed per peptide versus the standard of three. Prior knowledge of the composition of the raw proteins is an important asset for better identification of peptide sequences.

## 1. Introduction

Collagens are highly post translationally modified proteins found in animals (mainly mammals) and in humans. Native collagen is insoluble, while gelatine, corresponding to the extraction product of collagen from the bones and skins, is a hydrolysed form of collagen used in food due to its functional versatility. Prior to being used as sources of potential bioactive peptides, food proteins need to be submitted to hydrolysis, either mimicking physiological gastrointestinal digestion (e.g., simulated gastrointestinal digestion (SGID)) or industrial food-grade proteolytic enzymes, or both [1,2,3,4,5]). However, due to the low cleavage specificity of physiological and food-grade enzymes, qualitative and quantitative heterogeneities of peptides are generally important. Therefore, the most exhaustive identification of peptides is needed for adequate research of potential bioactive peptides (Figure 1).

This emergence of new health applications for food-derived peptides has led to the application of clinical proteomics/peptidomics to the agrofood field to characterize, as much as possible, the peptide population of complex mixtures resulting from the non-specific enzymatic cleavage of food proteins. The analytical method of choice is undoubtedly high-pressure liquid chromatography (HPLC), coupled with tandem mass spectrometry (MS/MS), which generates a very large amount of experimental data [6,7]. The latter directly depends on the separation capability (e.g., normal-flow versus nano-flow HPLC) of the chromatographic device used and the performance of the mass spectrometer used [8]. Presently, irrespective of the HPLC-MS/MS analytical combination used, the volume of experimental data is so extensive that the use of bioinformatics software is mandatory for processing the MS data [9,10].

In the past 20 years, concomitant to the establishment and development of web-based protein databases, commercial and academic bioinformatics software have been developed to manage these experimental MS data in order to identify the peptides, as well as their post-translational modifications (PTMs) [11]. Among these, PEAKS^®^ studio is a commercial software allowing peptide and PTM identification by comparison of peptide fragmentation spectral data, obtained by MS/MS versus targeted databases or directly by de novo sequencing. The PEAKS^®^ studio general parameters comprise several enzymatic cleavage sites and more than 300 types of PTMs [11]. Unfortunately, the setting of the nature of cleavage sites and the types of PTMs are constrained. For clinical peptidomics, the PEAKS^®^ studio search parameters related to the specificity of cleavage sites and the number of PTMs per peptide are set, by default, as “trypsin” and “a maximum of three PTMs per peptide”, respectively. Several features of the identified peptides, such as the peptide identification score, the protein score, the sequence coverage, the number of peptides, the parent proteins, the presence or absence of PTMs, the nature of the PTMs, etc., are provided by the software. However, an inappropriate setting of the search parameters induces, de facto, a high risk of missing numerous peptides. The challenge lies in the fact that no proteins are the same. Depending on their features, the identification of the generated peptides can, therefore, be compromised by intrinsic composition. 

In order to counteract the lack of peptide identification and their PTMs caused by software default setting, this study focused on optimizing peptide identification and their PTMs by considering the protein source and protein database-referenced knowledge. Beef, pork, and fish gelatines and their industrial gelatine hydrolysates before and after SGID, MS-grade trypsin (Tryp) digestion, or both (SGID/Tryp) were used in this study (Figure 1). Irrespective of the species, the difficulties with the collagen or gelatine polypeptides are the frequency of the main specific PTM (proline hydroxylation) and the site occupation yield, the lack of protein database annotation, and the generation of small peptides (di- and tripeptides) after SGID.

## 2. Materials and Methods

### 2.1. Materials

Gelatines and gelatine hydrolysates were obtained from Rousselot BV (Ghent, Belgium). The mass spectrometry grade trypsin/Lys-C mix was purchased from Promega (Madison, WI, USA). All other reagents were mass spectrometry-grade.

### 2.2. Preparation of Gelatine Hydrolysates

Gelatine hydrolysates were solubilized at a concentration of 10 mg·mL^−1^ in a solution of LC-MS-grade water and centrifuged for 10 min at 8000× *g*. The resulting pellets were removed, and the supernatants were subjected to peptidomics analysis.

### 2.3. Preparation of In Vitro Human Simulated Gastrointestinal Digestion (SGID) Samples

The in vitro SGID was adapted from the INFOGEST consensual static in vitro human gastrointestinal digestion protocol [12] and adjusted to the requirements of the study [13]. Three fluids were prepared to mimic the three first steps of gastrointestinal digestion: the oral, gastric, and intestinal phases. The mineral compositions of each fluid are listed in Table 1, and the pH of the solutions was adjusted to physiologically relevant values using solutions of NaOH (5 M) or HCl (5 M).

The entire digestion process was performed in a 200 mL reactor at 37 °C under constant stirring with a magnetic stirrer for 240 min. Two grams of protein sample (gelatines and gelatine hydrolysates from beef, pork, and fish) was added to the reactor, solubilized in 8 mL of Milli-Q^®^ water, and 8 mL of salivary fluid at pH 7 was added to reach a final protein/peptide concentration of 125 mg·mL^−1^. An aliquot of 4 mL was withdrawn at the end of the salivary step and stored at −20 °C. Then, 12 mL of gastric fluid containing pepsin at 6500 U·mL^−1^ was added, and the pH of the solution was adjusted to 2.5–3.0 by the addition of HCl. The final protein concentration during this step was 62.5 mg·mL^−1^. After 2 h, an aliquot of 4 mL was withdrawn at the end of the gastric step and stored under the same conditions. A 20 mL sample of intestinal fluid containing pancreatin at 45 U·mL^−1^ (based on trypsin activity) was added, and the pH of the solution was adjusted to 7 using a solution of NaOH. Intestinal digestion was carried out at a final concentration of 31.25 mg·mL^−1^ for 2 h, and an aliquot of 40 mL was withdrawn at the end of the intestinal step and stored. 

After being heated to 95 °C for 10 min, all samples were centrifuged at 13,400× *g* for 10 min at room temperature. The resulting pellets were removed, and the supernatants were diluted to 10 mg·mL^−1^ in a solution of LC-MS-grade water and then submitted to peptidomics analysis.

### 2.4. Preparation of Trypsinized Samples (Tryp or SGID/Tryp)

Samples that did not undergo the in vitro SGID protocol were solubilized in LC-MS-grade water to a concentration of 31.25 mg·mL^−1^ (corresponding to the final concentration after SGID). These samples and the SGID samples were heated to 50 °C for 10 min before incubation at 80 °C for 5 min in the presence of 25 mM ammonium bicarbonate and 5 mM dithiothreitol and then incubated at room temperature for 20 min in the dark in the presence of 10 mM iodoacetamide. MS-grade tryptic digestion (Tryp) was carried out after the addition of a trypsin/Lys-C mixture at 3 ng.µL^−1^ for 3 h at 37 °C, and again for 16 h at 37 °C, and then centrifuged for 10 min at 8000× *g*. The resulting pellets were removed, and the supernatants were diluted to 10 mg·mL^−1^ in a solution of LC-MS-grade water and finally submitted to peptidomics analysis.

### 2.5. Peptide Identification by RP-HPLC-MS/MS and Database Search

Aliquots of 10 µL of raw material, as well as Tryp, SGID, and SGID/Tryp hydrolysates, were chromatographically separated on an ACQUITY UPLC system (Waters, Manchester, UK) using a C18AQ column (150 × 3.0 mm, 2.6 µm, Interchim, Montluçon, France). Briefly, eluent A was ultrapure H2O containing formic acid (0.1%, *v/v*), and eluent B was acetonitrile (ACN) containing formic acid (0.1%, *v/v*). The ACN gradient (flow rate 0.5 mL.min^−1^) was as follows: 1% eluent B for 3 min, 1% to 30% eluent B over 45 min, 30% to 95% over 9 min, and then 95% to 1% eluent B over 3 min. The eluate was electrosprayed with the electrospray ionization source of the Q-TOF Synapt G2-Si™ (Waters) device. The MS analysis was performed in sensitivity, positive ion, and data-dependent acquisition (DDA) modes. The source temperature was set at 150 °C, and the capillary and cone voltages were set to 3000 V and 60 V, respectively. The MS data were collected for *m/z* values in the range of 50 Da and 2000 Da with a scan time of 0.2 s and a lock mass correction of 556.632 *m/z*, corresponding to singly charged leucine enkephalin. A maximum of 10 precursor ions were chosen for the MS/MS analysis, with an intensity threshold of 10,000. The MS/MS data were collected using the collision-induced dissociation (CID) mode and a scan time of 0.1 s at an energy collision of 8 V to 9 V for low *m/z* and a range of 40 V to 90 V for high *m/z*. All peptidomic analyses were performed in triplicates.

Database searches were performed using PEAKS^®^ Studio X+ software (Bioinformatics Solutions Inc., Waterloo, Canada) using the UniProtKB/Swiss-Prot databases restricted to *Bos taurus* (accessed March 2021—46,766 entries), *Sus scrofa* (accessed March 2021—120,911 entries), or *Oreochromis niloticus* (accessed July 2021—75,940 entires). A mass tolerance of 35 ppm and a MS/MS tolerance of 0.2 Da were allowed. The data searches were performed, specifying trypsin as the enzyme, and three missed cleavage sites were allowed (for Tryp samples) or without specifying an enzyme (for other samples). Two types of searches were carried out: (1) variable methionine oxidation was considered, with a maximum of three PTMs allowed per peptide, or (2) variable methionine oxidation and hydroxyproline were considered, with a maximum of six post-translational modifications allowed per peptide. The relevance of the peptide identities was assessed according to their identification scores provided by PEAKS^®^ Studio X+ using a *p*-value of 0.05 (*p* < 0.05) and a false discovery rate (FDR) < 1%.

The mass spectrometry proteomics data have been deposited with the ProteomeXchange Consortium via the PRIDE [14] partner repository with the dataset identifier PXD040820. To facilitate the depositing and the research of data, the following abbreviations have been assigned to the various deposited files: “G” for gelatine, “GH” for gelatine hydrolysate, “SGID” for simulated gastrointestinal digestion, “T” for trypsinized, “Conventional” for commonly-used bioinformatics search parameters (a maximum of three PTMs allowed per peptide), “Optimized” for optimized bioinformatics search parameters (variable hydroxyprolines and a maximum six PTMs allowed per peptide), and “Trypsin” for peptide identification performed denoting the choice of trypsin as the enzyme (“no enzyme” for the others files).

Moreover, in order to determine the peptide abundance, the peptide identification data were exported from PEAKS^®^ Studio X+ to a customized Microsoft Excel spreadsheet to generate heat maps of the amino acid occurrences (the number of times a given amino acid was found in an identified peptide) for the collagen alpha-1 chains of *Bos taurus* (CO1A1_BOVIN), *Sus scrofa* (A0A5G2QQE9_PIG), and *Oreochromis niloticus* (G9M6I5_ORENI). 

### 2.6. Statistical Analysis of the MS and MS/MS Data

A two-way ANOVA multiple comparison using a *p*-value less than 0.05 (*p* < 0.05) was carried out with the MS data and performed in duplicate.

## 3. Results

### 3.1. Mass Spectrometry Data

The similarities of MS and MS/MS data from different analytical runs have been evaluated as a result of the numbers of MS and MS/MS scans performed during the HPLC-MS/MS runs. For each species, Figure 2 provides the average number of MS and MS/MS scans and the corresponding standard deviations (SD) obtained from the triplicate HLPC-MS/MS runs. Surprisingly, the number of MS scans, as well as MS/MS scans, for all the proteolysed beef gelatine samples are close together. Indeed, on average, 7274 (±162) MS scans and 7741 (±449) MS/MS scans were recorded, irrespective of the hydrolysates analysed. On the other hand, the average number of MS and MS/MS scans recorded from RP-HPLC-MS/MS runs of proteolysed pork gelatine samples were 3914 (±374) and 13,494 (±568), respectively. Conversely, 9080 (±125) MS scans and 3664 (±206) MS/MS scans were recorded from the RP-HPLC-MS/MS runs of proteolysed fish gelatine samples. The two-way ANOVA multiple comparisons, using a *p*-value of less than 0.05 (*p* < 0.05), with the numbers (±SD) of MS and MS/MS scans demonstrated that, irrespective of the species, there was not a statistically significant difference between these numbers (±SD) with respect to the different digestion conditions. 

### 3.2. Impact of the Software Setting of the Enzyme Specificity on the Peptide Identification 

In the context of agrofoods, we have evaluated the impact of the software setting, and especially the specification of the cleavage specificity of the enzyme used versus the specification of the absence of a defined enzyme (no enzyme) on the number and nature of the identified peptides. This evaluation was only conducted for the MS-grade trypsin hydrolysates, since the gastrointestinal enzymes used are not specific for one or two strict proteolytic cleavage sites. 

As illustrated in Figure 3, irrespective of the enzyme setting, the bioinformatics retreatment of the MS/MS data led to a higher number of identified peptides (hatched- and non-hatched blue colour) for beef, pork, and fish gelatine samples hydrolysed by MS-grade trypsin. In the same manner, irrespective of the species, the MS/MS data, obtained for the trypsin-hydrolysed gelatine hydrolysates, always yielded more identified peptides when “Trypsin” was specified in the software as the specific enzyme experimentally used (hatched blue colour), rather than “no enzyme”. Indeed, the number of identified peptides for beef gelatine trypsin-based proteolysis was close to double: almost 492 versus 291 identified peptides, with the trypsin setting versus no specification of the enzyme, respectively. These numbers were 346 versus 167 for the pork gelatine trypsin-based proteolysis, and 273 versus 87 for the fish gelatine trypsin-based proteolysis. Blue Venn diagrams show that approximately 45% of the identified beef and pork peptides were identical irrespective of the setting used, while this percentage decreased to 30% for the fish samples. Consequently, when trypsin is used as the enzyme setting, the software yielded approximately 46%, 53%, and 68% of identified peptides for beef, pork, and fish, respectively. 

Conversely, the software setting related to selection of the enzyme used had no significant impact on the number of identified peptides from the MS/MS data obtained for the trypsin-proteolysed gelatine hydrolysates. Indeed, irrespective of the species, the numbers of identified peptides were close to 150, 235, and 125, respectively, for gelatine and gelatine hydrolysates (hatched and non-hatched red colour). However, the red Venn diagrams highlight that, irrespective of the species, an average of 40% of the identified peptides were identical for both settings, while 25% of the identified peptides differed from one setting to another. This percentage reached approximately 30% for the fish hydrolysates. Note that, when “trypsin” was specified, the peptide identification scores were 1.5 times higher for the beef hydrolysates, but not for the other species (data not shown). 

### 3.3. Impact of the Software Settings Related to the Number Allowed and the Nature of Specific PTMs on the Peptide Identification

Gelatine is a highly post-translationally modified protein, with the hydroxylation of proline (hydroxyproline, Hyp) being the most frequent modification [15,16,17,18,19]. Therefore, the peptides generated by proteolysis can encompass one to several hydroxyprolines, making the peptide identification against protein databases challenging. The assessment of the impact of software settings related to the number and nature of PTMs have been assessed using the same MS/MS data set as in Section 3.2 (analytical triplicates) for beef, pork, and fish, with (i) either commonly used bioinformatics search parameters (a maximum of three PTMs allowed per peptide or (ii) optimized bioinformatics search parameters (variable hydroxyproline and a maximum of six PTMs allowed per peptide). As illustrated in Figure 4, irrespective of the species or the proteolysis performed, the histogram bars show that the number of identified unique peptide sequences was often two times lower when using conventional bioinformatics search parameters than with optimized bioinformatics parameters. Obviously, in our case, the bioinformatic retreatment time is globally 1.6 to 3.4 more longer for the optimized search, but this, of course, depends on the size of (i) the protein database and ii) the MS and MS/MS data. For example, irrespective of the species, the number of identified peptides from trypsin-proteolysed gelatine (dark blue bar) was always higher than that for the other proteolysates, with 492, 346, and 273 versus 1063, 1216, and 658 identified peptides for the beef, pork, and fish, respectively. These values decreased greatly either when the degree of proteolysis was higher than that obtained with trypsin, as for proteolysis by SGID or SGID/Tryp (other blue and red histogram bars), or because “no enzyme” was used as the setting. Indeed, for the beef proteolysates, the number of identified peptides was close to 142 ± 14 when conventional bioinformatics search parameters were used and increased to 396 ± 61 when optimized bioinformatics search parameters were used. For the fish proteolysates, very much the same observation could be made, but the number of identified peptides was closer to 56 ± 12 than 142 ± 14, with an exception (a value of 126) for the trypsin-proteolysed gelatine hydrolysate (light red colour) when conventional bioinformatics search parameters were used. This number of 56 ± 12 increased to an average of 218 ± 31 identified peptides (always with an exception (a value of 416) for the trypsin-proteolysed gelatine hydrolysate) when optimized bioinformatics search parameters were used. Interestingly, irrespective of the proteolysis performed (except trypsin-proteolysed gelatine), the pork proteolysates always displayed a higher number (on average 249 ± 54) of identified peptides than the protein proteolysates of the other species. This average value increased to 630 ± 79 when optimized bioinformatics searches were performed. Remarkedly, irrespective of the species and the type of proteolysis, the number of identified peptides, bearing one or more hydroxyprolines (upper part of the histogram bars), was much higher when the optimized bioinformatics search parameters were used. This was the case for the beef, pork, and fish trypsin-proteolysed gelatines (dark blue bars). Indeed, for the trypsin-proteolysed beef gelatine, 121 peptides bearing one or more Hyp were identified when conventional bioinformatics search parameters were used, while 788 were identified using the optimized bioinformatics search parameters. These values for the trypsin-proteolysed gelatines from pork and fish ranged from 113 up to 1033 and 95 up to 518, respectively. This observation was also true for all other proteolysates, irrespective of their nature (SGID- and SGID/Tryp-proteolysed gelatines, gelatine hydrolysate, trypsin-, SGID-, and SGID/Tryp-proteolysed gelatine hydrolysates) (Appendix A presents details regarding the number of identified peptides) (the data sets generated and/or analyzed during this study are not publicly available due to company confidentiality, but they are available from elien.gevaert@rousselot.com upon reasonable request”.).

### 3.4. Impact of the Software Setting, This Is Related to the Number Allowed and the Nature of Specific PTMs on the Identified Peptides and the Identified Hyp-Bearing Peptides of the Protein COL1-A1

Irrespective of the animal origin (beef, pork, or fish), collagen 1 (COL1) is the most abundant protein of gelatine produced by the agrofood industry. Of the three chains of COL1, we have chosen the alpha-1 (A1) chain, which displays the best identification score, to illustrate, through a series of heat maps (Figure 5), for each animal species, the impact of the software setting related to the number and nature of PTMs with (i) either commonly used bioinformatics search parameters (a maximum of three PTMs allowed per peptide—Figure 5A,C,E) or (ii) optimized-bioinformatics search parameters (variable hydroxyproline and a maximum of six PTMs allowed per peptide—Figure 5B,D,F). The beef, pork, and fish proteolysates correspond to Figure 5A,B, Figure 5C,D, Figure 5E,F, respectively. Note that the protein sequences of the species displayed in Figure 5 are not aligned with each other. The number on the right corresponds to the number of COL1_A1-identified peptides. The known hydroxyproline position on the amino acid backbone of beef COL1_A1 is depicted as a thin vertical line at the top of the heat maps (Figure 4). Note that the hydroxyproline position of pork and fish COL1_A1 is not referenced in UniProt. Each heat map must be compared two-by-two (A-B, C-D, and E-F) to assess the impact of the conventional bioinformatics search parameters (A, C, and E) and the optimized bioinformatics search parameters (B, D, and F) on the identified peptides of COL1_A1. 

Overall, Figure 5B,D,F show (by the density of the red, orange, and yellow colours) that, irrespective of the proteolysis performed and the animal species, the optimized bioinformatics searches yielded more peptide identifications than the conventional ones, leading to a marked increase in the percentage of sequence coverage, as well as the characterization and location of Hyp on the protein backbone. These facts stand out for the beef and pork proteolysates (note the higher number of peptides identified in pork proteolysates), and are even more pronounced for the fish proteolysates. Indeed, for the latter, very few peptides (2 to 96 identified peptides) were identified with conventional bioinformatics search parameters in contrast to the optimized bioinformatics search parameters (54 to 237 identified peptides). Moreover, irrespective of the animal species, and as expected, the greater the degree of proteolysis (Tryp versus SGID or SGID/Tryp), the lower the number of identified peptides, irrespective of the search parameters used. Of note, irrespective of the search parameters used, the numbers of identified peptides were similar between SGID- and SGID/Tryp-proteolysed gelatine and SGID- and SGID/Tryp-proteolysed gelatine hydrolysates, thus demonstrating that MS-grade trypsin used after SGID generates few additional peptides and suggesting that SGID generates the vast majority of small-sized peptides.

From all of the MS data gathered from all beef gelatine proteolysate analyses, the optimized bioinformatics search parameters resulted in the potential identification and location of almost all UniProt-referenced Hyp (Appendix A) in the peptides (104 Hyp compared to 111 referenced in UniProt), especially when beef gelatine was hydrolysed by trypsin (89 Hyp compared to 111 referenced in UniProt (data not shown)). Among the 104 potential Hyp-bearing peptides identified, 24 were identified one or several times (between two and eleven times), with a PTM identification score of 1000 (Appendix A), and 21 of them are referenced in UniProt (P02453-CO1A1_BOVIN). Their positions on the protein backbone are depicted with red bars (Figure 5B).

In the same manner, taken together, the bioinformatic retreatment of the MS data for pork and fish gelatine and pork and fish gelatine hydrolysates using optimized bioinformatics search parameters resulted in the potential identification and location of 208 Hyp for pork COL1_A1 and 119 Hyp for fish COL1_A1 (Appendix A). Among these 208 and 119 potential Hyp-bearing peptides identified, 36 and 29, respectively, were identified one or several times (between two and fourteen times), with a PTM identification score of 1000 (Appendix A). Their positions on the protein backbones are depicted with red bars (Figure 5D,F). Again, for pork and fish COL1_A1, the positions of Hyp in the protein sequence are not referenced in UniProt. Finally, it should be noted that the identification and location of hydroxyprolines remain potential due to the MS fragmentation method used and will need to be confirmed by the use of fragmentation methods such as electron-transfer dissociation or electron-capture dissociation.

## 4. Discussion

In the data-dependent acquisition mode, the numbers of MS and MS/MS scans reflect the intensity of total ion current (data not shown) and the peptide heterogeneity of the various samples analysed. Indeed, since the instrumental scan time is fixed and invariable, the time to acquire the MS/MS data directly impacts the time remaining to acquire the MS data [20,21,22]. In other words, a high number of MS/MS scans is associated with a low number of MS scans (and conversely), indicating the presence of numerous peptide ions submitted to MS/MS fragmentation and, therefore, the richness of the MS/MS data for peptide identification. Moreover, convenient, adequate, and unbiased comparisons between MS and MS/MS data from different analytical runs require that the numbers of MS and MS/MS scans must not be statistically different [22,23]. As illustrated in Figure 2, the recorded MS and MS/MS data were fully comparable and, therefore, the conclusions were unbiased. Note that the numbers of MS and MS/MS scans recorded for the proteolysed beef gelatine and the proteolysed beef gelatine hydrolysates had almost the same value, while those corresponding to the pork and fish counterparts were inversely proportional. Indeed, when the number of MS scans is low, the number of MS/MS scans is high, and the converse is also true. The fact that a lower number of MS scans and a higher number of MS/MS scans were registered during the RP-HPLC-MS/MS runs of proteolysed pork gelatines and proteolysed pork gelatine hydrolysates suggests that the number, the intensity, or both of the peptide ions chosen for fragmentation in these samples are greater than in the other. The opposite was true for the proteolysed fish gelatine samples. 

To identify proteins, proteomics uses MS-grade trypsin hydrolysis due to the advantages associated with the hydrolysis of proteins by high-grade trypsin for RP-HPLC-MS/MS analysis: (i) the high specificity of cleavage by trypsin leads to a limited number of non-tryptic peptides, (ii) the C-terminal cleavage of proteins after a lysine or an arginine residue generates peptides displaying high protonic affinity (due to basicity of these amino acid residues), which facilitates the protonation of peptides and their subsequent detection and fragmentation, (iii) the molecular mass of the generated peptides is ideal for mass spectrometry due to, overall, the frequency and the position of lysine and arginine residues in the protein backbone, and (iv) the bioinformatics software takes advantage of the aforementioned points to manage the MS/MS data for peptide identification [24]. Therefore, the peptide identification is maximized and, consequently, also, the protein identification. This maximized peptide identification through conventional proteomics approaches is clearly confirmed by Figure 3, whereby, irrespective of the species, the MS/MS data corresponding to trypsin-hydrolysed gelatine led to a higher number of identified peptides when “trypsin” was selected in the software setting as the enzyme used. Surprisingly, when “no enzyme” was selected, the same MS/MS data were obtained, irrespective of the species, with twice as few identified peptides, thus demonstrating the impact of the software setting, regardless of the enzyme used. Therefore, it would be incorrect to think that the non-selection of the enzyme used by choosing “no enzyme” will take more time to be performed but will yield a higher number of identified peptides, since all the possible theoretical peptides derived from the amino acid sequences of the selected protein database will be compared to the MS/MS data. Aside from this, this enzyme setting (“trypsin” versus “no enzyme”) in the software has nearly no impact, irrespective of the species, on the number of identified peptides when the sample is a gelatine hydrolysate generated by enzymes displaying no strict specificity of cleavage sites and even when a secondary tryptic hydrolysis is performed on the gelatine hydrolysate. This point is of importance in the agrofood industry, since it largely uses enzymes with broad cleavage specificity to valorise protein-rich by-products. Irrespective of the species, the Venn diagrams demonstrate that almost all identified peptides from trypsin-proteolysed gelatines with no enzyme as the setting were also identified with trypsin as the setting. This is not the case for trypsin-proteolysed gelatine hydrolysates, for which only approximately 50% of the peptides were in common. In other words, for the latter, 25% distinct peptides were identified using each enzyme setting. Finally, five, six and four collagen types were identified from beef, pork, and fish gelatines proteolyzed by trypsin, and the collagen type I alpha-1 chain was the predominant type (data not shown). 

The PTMs attached to peptides influence the bioinformatics-based peptide identification according to the number of PTMs carried and the size of the amino acid sequence [25]. The greater the number of PTMs, the more the peptide identification is biased (depending on the MS-MS fragmentation method used), and the shorter the size of the amino acid sequence, the less convenient the peptide identification is. Gelatine is a highly post-translationally modified protein, with the hydroxylation of proline being the most frequent. However, the Hyp content and their locations are variable [11], and consequently, irrespective of the species, the Hyp locations on the amino acid backbone are mainly referenced in UniProt under the annotation “by similarity”. Prediction software of Hyp positions has been developed [26,27], and iHyd-Pse-AAC software predicted the potential presence of 88, 92, and 85 Hyp on the matured amino acid sequence of beef (CO1A1_BOVIN), pork (A0A5G2QQE9_PIG), and fish (G9M6I5_ORENI), respectively. 

The molecular mass of Hyp is very close to those of Leu and Ile. However, Montgomery et al. have clearly demonstrated that high-resolution MS allows distinction of Hyp from Leu/Ile, even though the molecular mass difference is only 0.03638 Da [16]. Obviously, the peptide identification is all the easier as the hydroxylations (irrespective of the position of the hydroxyl group) of proline are referenced in the protein databases. This is the case for chicken COL1_A1 (P02457-CO1A1_CHICK), but only by similarity to the latter for bovine, pork, and fish COL1A1. However, Zhang et al. have demonstrated that the proline hydroxylation of the high homology pork a2 collagen peptide (I^795^-K^815^) is slightly different compared to that of the bovine peptide (T^795^-K^815^) [25].

Therefore, we have evaluated the impact of the PTM setting of Peak^®^ Studio X+ software on the number of identified peptides containing either Hyp or not. In the particular case of gelatine, irrespective of the species, the settings of commonly used bioinformatics search parameters (a maximum of three PTMs allowed per peptide) or optimized-bioinformatics search parameters (variable hydroxyprolines and a maximum of six PTMs allowed per peptide) drastically changed the number of identified peptides and the number of identified Hyp-composed peptides. Note that the Peaks^®^ Studio’s queries launched with the optimized-bioinformatics search parameters required longer time to be over, but do not crash, probably due to the specific workflow (composed of successive search modules) of Peaks^®^ Studio compared to other proteomics/peptidomics softwares. For most proteolysates, the number of identified peptides was more than doubled when the software parameters were optimized. These results are consistent with the studies, dedicated to bovine placental α1(V)-collagen, of Yang et al., which highlighted the presence of a relatively large number of 3-hydroxyproline sites (approximately 10%) with less than 100% occupancy [15], as well as other studies carried out at the same time [28,29,30]. 

Concomitantly, the number of Hyp identified in the amino acid sequence of identified peptides was also doubled using the optimized parameters. Therefore, almost all (104 out of 111) predicted Hyp of beef COL1_A1 referenced in the UniProt database have been potentially re-identified using the optimized bioinformatics search parameters to retreat the MS data from proteolysed gelatines and proteolysed gelatine hydrolysates. Moreover, of 24 Hyp-bearing peptides identified with a high degree of certitude (with a PTM identification score of 1000) regarding the presence and location of Hyp, 21 of them are referenced in UniProt. Concomitantly, this approach identified 208 and 119 potential Hyp in the pork and fish COL1_A1 backbones, respectively. Among these, 36 and 29 Hyp-bearing peptides were identified with a high confidence score (PTM identification score of 1000). Irrespective of the animal species, their repartition was homogeneous on the protein backbone, and there were Hyp position homologies with beef-COL1_A1 (Appendix A).

## 5. Conclusions

This study, performed with proteolysed gelatines and proteolysed-gelatine hydrolysates of beef, pork, and fish, demonstrates the impact of the MS data retreatment and the bioinformatic search parameters on the number and the nature of the identified peptides. When the enzyme used is specific to one or two cleavage sites, such as in case of trypsin, the setting of the PEAKS^®^ Studio X+ (a commercial bioinformatics software dedicated to peptidomics and proteomics) related to the enzyme selection is crucial to optimize the number and the nature of the identified peptides from these proteolysates. Surprisingly, irrespective of the animal species, when “no enzyme” was selected, the same MS/MS data yielded two times fewer identified peptides, thus demonstrating the impact of the software setting regardless of the enzyme used. Therefore, it seems wrong to think that the non-selection of the enzyme experimentally used will yield the same or a higher number of identified peptides when the proteolysis is performed by a specific enzyme, such as trypsin. Conversely, irrespective of the species, the enzyme setting (“trypsin” versus “no enzyme”) has nearly no impact on the number of identified peptides when the gelatine proteolysates resulted from enzymes displaying high cleavage site specificities. This point is of paramount importance in the agrofood industry, since it largely uses enzymes with broad cleavage specificities to valorise protein-rich by-products. 

In the same manner, the PTM setting in PEAKS^®^ Studio X+ greatly impacts the number and the nature of the identified peptides. Indeed, optimized parameters (variable hydroxyprolines and a maximum of six PTMs allowed per peptide) yielded two times higher identified peptides compared to the conventional setting (a maximum of three PTMs allowed per peptide). Concomitantly, the number of potentially Hyp-containing peptides was also greatly enhanced when the optimized parameters relating to PTMs were selected to carry out the peptide identification from MS data of proteolysed gelatines and proteolysed gelatine hydrolysates. As a result of this optimization of MS data bioinformatic retreatment, this study experimentally highlighted the potential presence of 104 Hyp in the amino acid sequence of bovine COL1A1 (P02453-CO1A1_BOVINE, where 111 are referenced by similarity in UniProt), and it identified the number and location of potential Hyp in the maturated amino acid sequence of A0A5G2QQE9_PIG and G9M6I5_ORENI. Overall, 24, 36, and 29 Hyp were identified in peptides with a high confidence score for the beef collagen alpha-1(I) chain, and its pork and fish counterparts, respectively. Among the bovine ones, 87.5% (21 out of 24) are referenced in UniProt, suggesting a good correlation of our MS data with the published literature. It should be noted that all these Hyp have a homogeneous distribution throughout the amino acid sequence of the above-mentioned maturated proteins (Appendix A). Although, in this study, the potential presence of Hyp relied on the use of high-resolution mass spectrometry, the experimental confirmation (based on the characteristic MS/MS fragment for the presence of Hyp in an amino acid sequence) will need to be confirmed using a more informative MS/MS method. 

## Figures and Tables

**Figure 1 foods-12-02524-f001:**
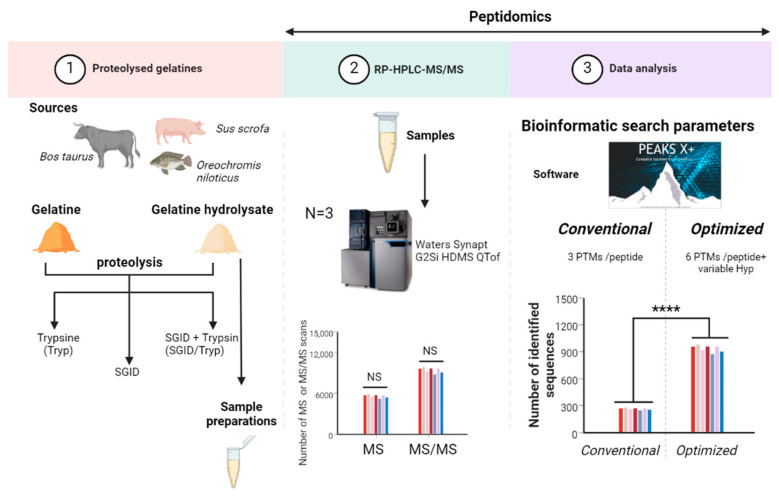
Workflow of the strategy. (1) Beef, pork, and fish gelatines and their industrial gelatine hydrolysates before and after MS-grade trypsin (Tryp) digestion, simulated gastrointestinal digestion (SGID), or both (SGID/Tryp), which were used to perform (2) RP-HPLC-MS/MS analysis. (3) Bioinformatics treatment was evaluated using the commonly used bioinformatics search parameters (a maximum of three post-translational modifications (PTMs) allowed per peptide) or optimized bioinformatics search parameters (variable hydroxyprolines and a maximum of six PTMs allowed per peptide). ****, significant statistical differences.

**Figure 2 foods-12-02524-f002:**
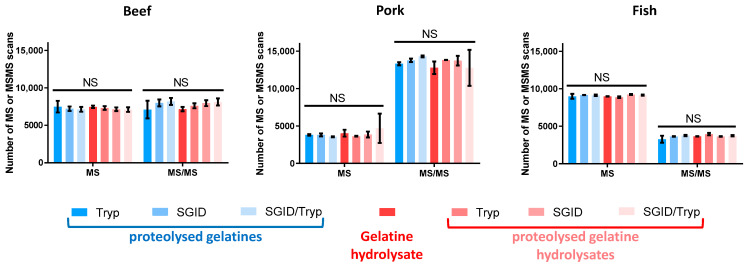
Mass spectrometry data. Number of MS and MS/MS scans performed during the RP-HPLC-MS/MS runs of (i) proteolysed gelatines, (ii) the gelatine hydrolysate, and (iii) proteolysed gelatine hydrolysates from beef, pork, and fish. The proteolyses were carried out using (i) MS-grade trypsin (Tryp), (ii) simulated gastrointestinal digestion (SGID), or (iii) both (SGID/Tryp). All hydrolysates were analysed in triplicate. NS, not significant.

**Figure 3 foods-12-02524-f003:**
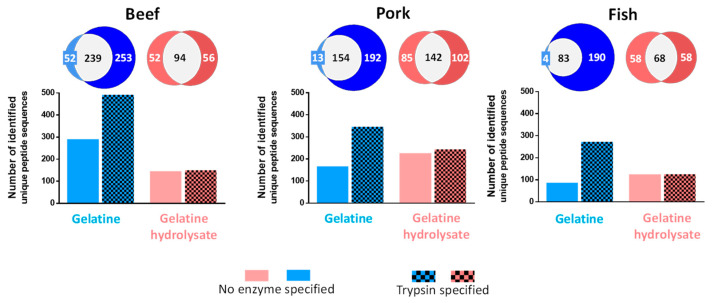
Impact of specification of the enzyme used on the number of identified peptides. Numbers of identified peptides (de novo peptide identification excluded) with Peaks^®^ Studio X+ software obtained after RP-HPLC-MS/MS analysis of gelatine and gelatine hydrolysates from beef, pork, and fish. The proteolyses were carried out using MS-grade trypsin (Tryp). All hydrolysates were analysed in triplicate, and the peptide identification was performed with a single query, combining the three replicates, specifying the choice of “no enzyme” (non-hatched) or “trypsin” (hatched) as the enzyme. The proportional Venn diagrams report the numbers of common and distinct identified peptides according to the software settings.

**Figure 4 foods-12-02524-f004:**
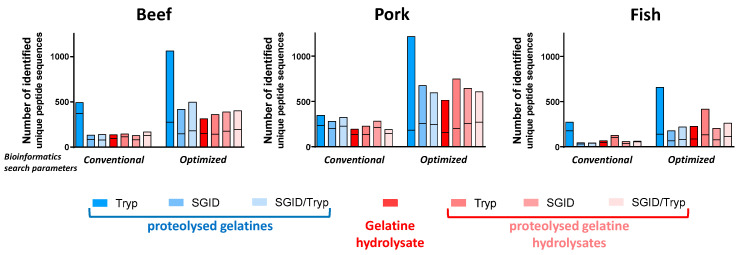
Total number of identified peptides using the commonly used bioinformatics search parameters (a maximum of three post-translational modifications (PTMs) allowed per peptide) or optimized bioinformatics search parameters (variable hydroxyprolines and a maximum of six PTMs allowed per peptide). The numbers of identified peptides with Peaks^®^ Studio X+ software obtained after RP-HPLC-MS/MS analysis of proteolysed gelatines, gelatine hydrolysate, and proteolysed gelatine hydrolysates from beef, pork, or fish. The proteolyses were carried out using (i) MS-grade trypsin (Tryp), (ii) simulated gastrointestinal digestion (SGID), or (iii) both (SGID/Tryp). For each histogram bar, the upper part corresponds to peptides bearing one or more hydroxyproline(s) and the lower part to other peptides. All hydrolysates were analysed in triplicate, and the peptide identification was performed with a single query combining the three replicates.

**Figure 5 foods-12-02524-f005:**
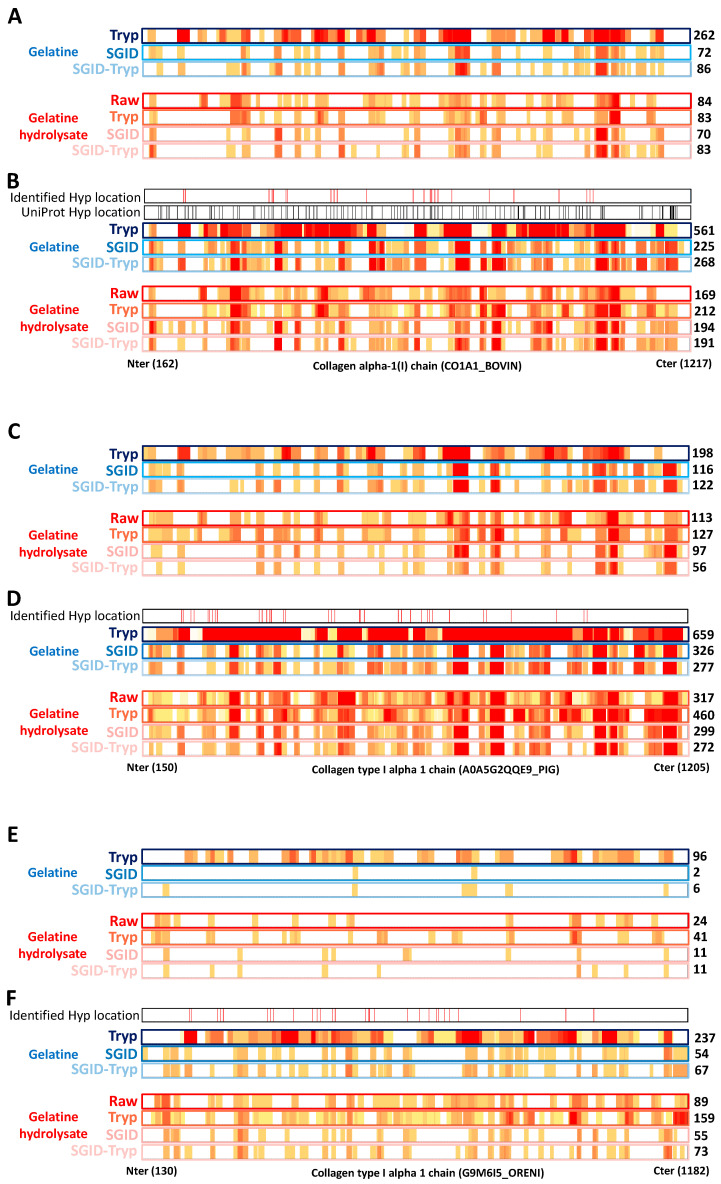
Peptide patterns (heat maps) showing amino acid occurrence in the identified peptides along the amino acid sequence of collagen type-I alpha-1 obtained using the commonly used bioinformatics search parameters (maximum of three PTMs allowed per peptide) (**A**,**C**,**E**) or optimized bioinformatic search parameters (variable hydroxyprolines and maximum of six PTMs allowed per peptide) (**B**,**D**,**F**). Note that the protein sequences of the species are not aligned with each other. The number at the right of each pattern corresponds to the number of identified peptides in the sample. Heat maps surrounded by a blue frame correspond to proteolysed gelatines, and heat maps surrounded by a red frame correspond to proteolysed gelatine hydrolysates from beef (**A**,**B**), pork (**C**,**D**), or fish (**E**,**F**) hydrolysed by (i) MS-grade trypsin (Tryp), (ii) simulated gastrointestinal digestion (SGID), or (iii) both (SGID/Tryp). For A and B, the line “hydroxyproline position” (black vertical lines) refers to the Hyp location extracted from UniProt. The Hyp locations in UniProt are referenced by similarity with Chicken COL1A1, but not referenced for pork and fish. The Hyp locations identified with a PTM identification score of 1000 are depicted above as red vertical lines. The UniProt accession numbers used are CO1A1_BOVIN, A0A5G2QQE9_PIG, and G9M6I5_ORENI for bovine, pork, and fish, respectively. The N-terminal pro-peptides and the C-telopeptides of pork and fish gelatines have been evaluated by BLAST alignment of the sequences below using the bovine one as the reference.

**Table 1 foods-12-02524-t001:** Chemical composition, protease activities, and pH used for the various compartments of the in vitro SGID.

Chemical composition	**Oral**	**Gastric**	**Intestinal**
	KCl (15.1 mM)	KCl (6.9 mM)	KCl (6.8 mM)
	KH_2_PO_4_ (3.7 mM)	KH_2_PO_4_ (0.9 mM)	KH_2_PO_4_ (0.8 mM)
	NaHCO_3_ (6.8 mM)	NaHCO_3_ (25 mM)	NaHCO_3_ (85 mM)
	MgCl_2_ (0.5 mM)	MgCl_2_ (0.1 mM)	MgCl_2_ (0.33 mM)
	NH_4_HCO_3_ (0.06 mM)	NH_4_HCO_3_ (0.5 mM)	NaCl (38.4 mM)
		NaCl (47.2 mM)	
Proteases		Pepsin 6500 U·mL^−1^	Pancreatin 45 U·mL^−1^
pH	7.0 ± 0.6	3.0 ± 0.6	7.0 ± 0.6

## Data Availability

The mass spectrometry proteomics data have been deposited with the ProteomeXchange Consortium via the PRIDE (Perez-Riverol et al., 2019) [14] partner repository with the dataset identifier PXD040820. The data sets generated and/or analyzed during this study are not publicly available due to company confidentiality, but they are available from elien.gevaert@rousselot.com upon reasonable request.

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
