# Peer review of "Impact of Bioinformatics Search Parameters for Peptides’ Identification and Their Post-Translational Modifications: A Case Study of Proteolysed Gelatines from Beef, Pork, and Fish"

_foods, 2023, doi:10.3390/foods12132524_

Round 1

Reviewer 1 Report

The authors compare standard and custom software settings for mass-spectrometric analysis of enzyme treated beef, pork, and fish samples.  They make two primary conclusions.  First, inclusion of a specific enzyme in the search parameters optimizes results.  This appears to be an expected result and should also conserve the time used for the search.  Two, the inclusion of six PTMs allowed per peptide (versus the standard 24 of three) also optimizes results for these sample materials.  This also appears to be an expected result, but would likely increase the time used for the search.  Finally, they conclude that prior knowledge of the sequence proteins is an important asset for improved identification of peptide sequences.  This is also an expected result.  The research, figures and text are well designed, straightforward, and easily interpretable. 

Minor points to be addressed include,  

1)      There are no error bars in figures 3 and 4

2)      How long were the search times between the standard and custom search?

3)      Can the customized excel spreadsheet be included in a supplement?

Author Response

Dear reviewer,

Thank you for your useful comments and suggestions that undeniably improve our manuscript. For clarity, you will find, in the attached file, our answers and the list of changes made in accordance to your major comments.

Best regards,

Dr. Christophe Flahaut

Reviewer 2 Report

In this manuscript, the authors focused their efforts in to analyze the effect of the search parameters settings of the proteomics software over the final lists of peptides/proteins obtained. More precisely, they compared the effect of two search parameters configurations (classical vs optimized) using the PEAKS X+ software on two different samples (gelatine vs gelatine hydrolysates) from three animal species (beef, pork and fish) digested with three different combination of enzymes (trypsin, simulated gastrointestinal digestion (SGID) and SGID+trypsin). The manuscript very carefully dissects all the differences found in all the samples and digestion conditions however, there are several issues that still remain unclear which must be solved:

1) Regarding to the search settings the authors indicated that the mass tolerance for the MS and MS/MS searches were 35 ppm and 0.2Da, respectively. If the authors are optimizing the search parameters, 35 ppm looks high for a high-resolution mass spectrometer (Synapt G2 SI). Indeed, for a 1000Da peptide, an error of 35ppm error is equal to 0.035Da which is more or less the mass difference between the Hyp and Leu/Ile (0.03638Da). In fact, the authors propose this problem as something that can be solved by the high-resolution mass spectrometers, however, in the step of searching the MS/MS spectra, the reduction of the error tolerance to 10-15 ppm, would reduce the theoretical peptides candidates and, subsequently, the errors in the peptide sequence assignments. In addition, the number of peptides with more than 10 ppm of error is quite low (only around 10.3% of the peptides shown, for instance, in the excel file "Beef_G_T_Optimized_Trypsinspecified_protein-peptides.csv" have more than 10ppm of error). If we assume that the error tolerance reduction will increase the peptides score most likely we will obtain a net gain of peptides in the final list of peptides after the filtering by the FDR.

2) The major issue of the manuscript is shown in figure 5. As pointed out by the authors, the gelatine hydrolysate from pork shows no peptides for Collagen type I alpha 1 chain when the conventional parameters were used. However, the same sample, i.e. the same RAW file, shows 317 peptides in the counterpart search with optimized parameters. 85 of these peptides are “unmodified” according to the “Pork_GH_Optimized_protein-peptides.csv” file therefore they must be detected in the classical search. Could the authors confirm that they are using a unique database for each specie? Could the authors please upload the databases into the repository so that we can reproduce the search? If zero peptides are the truth, then the authors have to explain the presence of 85 unmodified peptides in the optimized parameters. By contrast, if 317 peptides are the truth, then they have to explain the absence of unmodified peptides in the classical search.

Related to that, in the classical search (see for example the counterpart file for the optimized search described above which is called “Pork_GH_Classical_protein-peptides.csv”) there are peptides found by PEAKS when the “PEAKS PTM” search is used (see the column called “Found By”). PEAKS PTM” is a PEAKS software module that allows to search the raw files against all the PTMs defined in the software, regardless of the fixed/variable PTMs defined in the “PEAKS DB” search.  If the “PEAKS PTM” is used in the classical search then the peptides with three or less hydroxyprolines should be also found in this search. Could the authors explain, please, why they do not find hydroxyprolines in the classical search if they use the PEAKS PTM module?

3) The number of entries of the databases should be added to the material and methods section.

4) The number of MS vs MS/MS spectra in the three species is different as shown in figure 2. It should be interesting to compare the Total Ion Chromatogram intensities of all the samples to check whether this is the reason for this behaviour.

5) In figure 3 the authors show the impact of the specification of the enzyme on the number of identified peptides. All the hydrolysates were analyzed in triplicate but there are not the SD bars. In agreement to the figure 2, the SD bars should be added on this figure. 

Typo errors:

6) In the line 336 the statement "Overall, Figures 5B, D and E" should be modified by "Overall, Figures 5B, D, F.

7) In the line 402, the statement "As illustrated in Figure 1" should be modified by "As illustrated in Figure 2". 

8) in the conclusions section, line 510-512, the authors state that the optimized parameters give fewer identified peptide than the conventional settings. This is in opposition to what we observe in figure 4 and, in general, to the idea exposed in the manuscript. Please, correct this typo error. 

Author Response

(The authors gave the same response as above.)

Reviewer 3 Report

This manuscript describes a phenomenon that is quite well-known  in proteomics studies, though often neglected or underestimated. It is indeed more relevant to this type of highly PTM modified proteins (collagens) a.o. in food and bioactive peptide research. The impact of the choice of search parameters on the output is often overlooked. So it is relevant to bring it (again) under attention. 

The manuscript is well written, with clearly described set-up, method description and results description. There is a clear and strong focus on this specific aspect of digestive enzyme and PTM settings. However, broader implications of the subject matter are out-of -focus  of the manuscript.

 I have a few comments:

- some references on reviews of bioinforomatics piepelines is quite out-dated, : ref 6 and 10 could be replaced by more recent review on the topic
- in (especially) refernce 28 and 30 the major words are in {curly brackets}, this may be an editing fomrat, but should be checked

- I am slightly surprised that the search setting of non-specific enzyme and up to 6 variable modifications actually works on a complete proteome ( of either bovine, pork or fish). As the search space is gigantic. Many proteomics search engines will crash on memory issues with these settings. I suppose PEAKS has a particular two-step approach included in the workflow which in the first step identifies proteins without modifications, and in a second stage searches for potential modified peptides on the small subset of identified proteins. Normally , we would follow such two-step approach manually (with other software, like MaxQuant). 
I suggest to discuss this issue in the discussion section of the manuscript. 

- I am also surprised to see in the peptide-protein tables (as presented in PRIDE) that only a very small number of protein accessions (5-8) in 3-5 groups  have been identified. I would have expected also isoform variants of the collagen subunits. Apparently these are merged in the PEAKS software.  Is that the case? Some discussion on this may be appropriate.

- also other modifications have been detected in the data, but these are not at all mentioned or discussed.

Small edits:

- line 207: might be added : "with respect to the different digestion conditions"

- line 395: "time scan"should be  "scan time"

- line 434: delete "choose"

- line 512:  "two times fewer" should be "two times higher"  , I guess?

Author Response

(The authors gave the same response as above.)

Reviewer 4 Report

The paper is well written and authors clearly described the design and purpose of the work. Authors presented an interesting perspective to improve practical issues when it comes to translation of mass spectrometry-based protein/peptide identification to agro-food field. I have no improvements to suggest for the proteomic point of view. 

Author Response

(The authors gave the same response as above.)

Round 2

Reviewer 2 Report

The answers provided by the authors clearly solved the major issue found in the figure 5 of the paper. However, although the technical explanation provided by the Peaks Studio Support Team is clear, it seems that in this specific case, the problem turns around a problem in writing the protein-peptides.csv and peptide.csv files (by the way, it should be interesting to know the explanation from the Peaks developers, for this writing problem). It is known that changes in the search space (for example, increasing the number of variable modifications from 3 to 6) clearly alter the calculation of the FDR. However, it is difficult to accept that 85 peptides from one single protein drop as a consequence of these changes. If it is true, it could be an interesting exercise to check the dataset to look for more proteins with similar behavior.